# Thermal Analysis of Coupled Resonant Coils for an Electric Vehicle Wireless Charging System

Chunming Wen [1,2,*,†], Qing Xu [2,†], Minbo Chen [2], Zhanpeng Xiao [2], Jie Wen [2], Yunyun Luo [2], Xiaohui Zhao [2], Yuanxiong Liang [3] and Kairong Liang [4]

1  Guangxi Key Laboratory of Hybrid Computation and IC Design Analysis, Guangxi Minzu University, Nanning 530006, China
2  College of Electronic Information, Guangxi Minzu University, Nanning 530006, China; xuqing0202@stu.gxmzu.edu.cn (Q.X.); chenminbo0610@stu.gxmzu.edu.cn (M.C.); isxiaozhanpeng@163.com (Z.X.); wen10032@163.com (J.W.); lleybey@163.com (Y.L.); zhaoxiaohui586@163.com (X.Z.)
3  Guangxi Lanchuang New Energy Automotive Equipment Co., Ltd., Liuzhou 545000, China; gxfxyxny@163.com
4  Guangxi Automobile Group, Liuzhou 545000, China; qinyan02@wuling.con.cn
*  Correspondence: wenchunming@gxmzu.edu.cn
†  These authors contributed equally to this work and should be considered co-first authors.

**Abstract:** Electric vehicles use wireless energy transmission to obtain energy, which can effectively avoid the shortcomings of traditional methods. As the carrier of radio energy transmission and reception, the high temperature of the coil triggers the degradation of wireless transmission performance and the aging of the coil, which may cause fire and other safety problems in serious cases. This paper studied the temperature distribution of the magnetically coupled coil model for electric vehicles. Based on the study of the basic law of heat transfer, the coil model was established using ANSYS software, and the boundary conditions and relevant parameters were set. After many simulation experiments and comparisons, it was finally determined that the transmitting coil and the receiving coil were the same sizes, the inner diameter of the coil was 100 mm, the outer diameter of the coil was 181 mm, and the coupling distance between the transmitting coil and the receiving coil was set to 60 mm. Coil models were simulated and analyzed using different materials. The simulation results show that after 30 min of system operation, the material chosen from the temperature range may have been gold, silver, copper, or aluminum, but from the comprehensive consideration of cost and performance, the material of the coil in the model was finally set to copper. Copper was the best material; its temperature maximum was 74.952 °C and lower than the safety value of 80 °C. It is hoped that this study will provide a reference for wireless charging coil design.

**Keywords:** coupled coil; heat transfer; radio energy transfer; optimization; temperature field distribution

## 1. Introduction

Temperature, vibration, humidity, and other factors are the main causes of power electronics failure; relevant surveys show that 55% of these failures are caused by high operating temperatures [1]. In the current theory of electronic products, there exists the very famous "10 °C law", that is, in the case of temperature in units of 10 degrees Celsius, for each unit of temperature increase, the failure rate of the product will be greatly increased. Therefore, with the advantages of environmental protection, easy maintenance and no noise generation, electric vehicles are a major component of the future transportation system and can effectively reduce $CO_2$ gas emissions [2]. Electric vehicles are charged by wireless energy transmission, which can effectively solve the shortcomings of traditional methods, such as inconvenient carrying of cables and poor contact of plugging ports. The coil is

an important part of the wireless charging system. If the temperature of the coils is too high during operation, it will shorten the service life of the devices and cause fire and other safety problems in severe cases. The thermal stability of the device is important for the system to work safely and efficiently. Therefore, to address the thermal stability of the magnetically coupled coil of an electric vehicle, this paper develops a study of the temperature distribution of the modelled magnetically coupled coil of an electric vehicle. By analyzing the distribution of coil heat through thermal simulation, it was determined that it would be beneficial for the system to take effective cooling measures in order to efficiently perform wireless energy transmission.

Electronic products have more and more functions and stronger performance, and the number of devices placed on the circuit board is increasing in unit space. With the change in the working environment, the power loss of components in electronic products increases, and their calorific value is also constantly increasing, thus increasing the overall structural temperature of electronic components. Temperature is an important factor in the failure of electronic components. Temperature design of electronic products is needed to ensure that the product works in an operational temperature environment. In the process of designing electronic products, it is an important aspect of product design to improve the heat dissipation performance of products to ensure the long-term stable operation of wireless power transmission. The direct criterion for evaluating the heat dissipation performance is the working temperature of the product, so it is of great significance to design the heat dissipation system and reduce the working temperature of the product.

For the heat generated by electronic products, in this technology of controlling heat, international countries are more advanced. International countries began to study the problem of temperature rise in electronic equipment in the 1960s, and gradually developed the thermal analysis of electronic equipment, thermal design technology, and thermal testing technology. The thermal analysis technique, known as the thermal simulation analysis method, is based on the principle of using the finite element method combined with computer-aided technology to simulate and analyze the heat in the early stages of product design and to correct the problems in the design by computer-aided technology to finally obtain the optimal product design solution. In [3], Yasufuku reviewed the current status of the application of thermal analysis techniques in electrical and electronic material identification and quality control. Luis et al. used the boundary integral method to predict the temperature of each component on PCB boards [4]. Iwasaki et al. [5] addressed the increasing demand for electronic circuits with greater complexity, versatility, and reliability to meet these requirements, and high-density multi-chip packaging techniques were used. The density of devices on the circuit was higher, caused by increased power per unit area. Therefore, as it is important to conduct thermal analysis during the design of electronic circuits, the thermal network method was proposed to be applied in thermal analysis, and a simple model for multi-chip component analysis was proposed as a preliminary thermal analysis design tool. Based on the thermal analysis results, the effectiveness of the thermal network method and the simple thermal analysis model were verified. The literature [6] investigated the effect of the operating frequency of an electrically inductive charging system on heat generation and overall system transmission efficiency. This result is due to the change in material properties caused by temperature and frequency, and for a faster subsequent optimization, a step-down module was added to the system simulation, implemented to allow an accurate calculation of the losses in the different system parts. At 100 min, the temperature of the coil in the system was constant and had a value of 122.7 °C. A thermal study of a lightweight onboard receiver module for an electric vehicle wireless energy transmission system has been presented in [7]. To address the thermal management challenges that arose during the interplay between the high transmission power of the automotive wireless power transmission system and the multi-component assembly of the receiver module, the physical behavior of the lightweight concept was then proposed to be applied in a simulation model, which was verified experimentally with key active components. Transmission lines were arranged in a triangular shape for most of their paths

in [8], to mitigate the magnetic fields generated by underground transmission lines. Such an arrangement induced strong cable coupling, and the use of flat structures in the joint area can effectively reduce cable stresses. However, it introduces a new problem of magnetic field values exceeding the threshold value. To solve the problems caused by the structure, the High Magnetic Coupling Passive Loop (HMCPL) technique was introduced to reduce the magnetic field strength, which is based on the principle of generating currents of equal amplitude and opposite phase for each source conductor to counteract the magnetic field generated in a specific area. Since the space is small and the number of heat sources is doubled, the thermal behavior of steady-state and transient states under these conditions was studied. When the diameter of the heat source conductor was larger than 30 cm and smaller than 60 cm, the temperature drops sharply to 84.5 °C, which meets the reference standard. In [9], it was studied that the temperature inside the permanent magnet coupling increases during operation due to the presence of induced eddy currents, which affects the performance of the permanent magnet coupling. The temperature field simulation experiment used Workbench software, in which the maximum internal temperature was 59.104 °C and lays the foundation for the subsequent thermomagnetic coupling analysis of the permanent magnet coupling. In [10], in conducting simulations for power electronics applications, the losses of magnetic components cause significant temperature increases, which can lead to significant changes in the components, and for this reason a model was proposed that was a nonlinear model of hysteresis, electromagnetic winding, and thermal behavior in one model. Both measurements and simulations show that the method was accurate for both static thermal conditions and dynamic self-heating conditions in the thermal domain, between 27 and 154 °C, where the saturation flux density decreased and the coercivity decreased with increasing temperature. The literature [11] studied eddy current retarders in auxiliary braking devices for heavy vehicles, which experience local high temperatures during operation, leading to a decrease in braking power. Hence, a new type of integrated eddy current buffer was proposed, and its thermomagnetic coupling characteristics were studied for temperature field simulation experiments. The results show that the heat source was concentrated on the inner side of the eddy current buffer stator, which would reduce the braking torque. To reduce the temperature better, a liquid cooling channel near the inner side of the stator can effectively reduce the stator temperature and extend the braking time.

There have been several achievements of domestic research on the thermal analysis of electronic products using finite element techniques. Ying Liang et al. [12] proposed an optimization study of the chip thermal layout of stacked 3D multi-chip components based on the thermal superposition model to study the optimization of the chip thermal layout within the stacked 3D Multi-Chip Module (3D-MCM); based on the thermal superposition model and combined with the heat conduction equation, the temperature of the chip was selected as the evaluation index, the adaptation function was used for 3D-MCM thermal layout optimization, and the optimal chip thermal layout scheme was derived by using the genetic algorithm to optimize the chip thermal layout. Jia Songliang and Zhu Haoying et al. [13] investigated the thermal resistance Rwc of IC ceramic packages by the finite element analysis technique, which showed that finite element thermal analysis was a very useful method. In [14], a foreign object detection coil array that does not affect the power transmission capability of the wireless power transmission (WPT) system and has no blind spot for detection was proposed for the problem of metallic foreign objects falling into the charging area of electric vehicles, which affect the electrical characteristics of the system and generate eddy current heat. The effectiveness and feasibility of a variety of common metallic foreign body verification detection methods were verified by experiments. The literature [15] considered the influence of aging factors on the estimation of the structural temperature of the battery, and the finite element difference method was used to estimate the structural surface temperature based on the heat transfer law of electrochemistry. The highest values of the internal temperature of the battery were 28.15 and 32.58 °C when the discharge rate reached 0.5 C and 1 C. Multiple sets of temperature tests on

the internal battery under different discharge rate conditions were not performed. The literature [16] on the processing of long stainless-steel tubes in the process of magnetic field generator temperature rise was relatively large, more comprehensive than that on the allowable temperature of the winding, inducing magnetic honing process; the limited scope of this was a problem. To ensure the sustainability of the process, the cooling method of external circulation water cooling was proposed. Simulation analysis and test results show that when the magnetic field generator does not use a water-cooled circulation device, the temperature reached the system's maximum value of 75 °C. Under the same conditions, the temperature of the system dropped to 60.6 °C after adding the water-cooling circulation device. From the study, it was clear that the water-cooling method can reduce the temperature rise of the system and improve its stability. The disadvantage was that the rotor grinder speed in the magnetic honing system was not carried out in multiple gears for simulation comparison. The issue of temperature field distribution on the length of service life of ferric tanks was studied in [17]. This literature used finite element software to analyze the effect of factors such as the thickness of the refractory bricks on the distribution of the temperature field. After the finite element software simulation analysis, it was determined that the smaller the thickness of refractory bricks, the more the radial temperature gradient increases and the value of thermal conductivity of refractory bricks becomes larger, raising the temperature of the outer surface of the tank wall. A reasonable choice of the thickness of the refractory bricks can effectively reduce the temperature gradient and the surface temperature and extend the service life of the iron tank. In [18], travel temperature directly affects the specific energy consumption of electric vehicles through the interference of various aspects, which leads to differences in their charging power requirements under different circumstances. To solve the specific energy consumption problem, a charging power calculation model for electric vehicles that calculated the travel temperature is proposed, based on statistical principles. The model concludes that low temperatures affected the charging load of electric vehicles more than high temperatures. The object of study in [19] is the magnetic coupler in the wireless energy transfer system of electric vehicles, whose operated temperature determines the reliability and service life of the system. Firstly, the thermal circuit model is proposed for the magnetic coupler, the heat generation mechanism of each part in the magnetic coupler was analyzed, then the thermal power of each part was calculated, then the temperature distribution of each part was simulated, and finally, the physical prototype was built, and the experimental results are consistent with the theoretical simulation results. However, the amount of measured temperature data in the simulation experiment is relatively small. In [20] electric buses, due to Heating Ventilation and Air Conditioning (HVAC) systems, consume a great deal of energy, causing the electric bus operational journey reduction. Furthermore, the HVAC system on electric buses operates at a level higher than the passenger thermal comfort requirements, so it was proposed to study the effect of this system on energy loss and thermal comfort. An appropriate thermal comfort model was determined, and the required climate parameters were entered and measured by a self-developed sensor station, while for electric bus energy consumption, the battery charge status and available travel range were measured with an embedded data logger. The climate test was carried out by turning the heating on and off for the electric bus under winter conditions. The experimental results show that the energy consumption on the electric bus increased by a factor of 1.9 when the heating was turned on, with a corresponding decrease in the State of Charge (SoC) and travel range. In [21], a 15-kW wheeled motor was chosen as the object of study to analyze the coupling factors between the electromagnetic field, temperature field, and flow field; a three-field coupling model of the wheeled motor with some complexity was established and analyzed to verify the validity of the model. The maximum temperature of the wheeled motor was 150 °C after 140 min of operation in the driving condition, which does not exceed the insulation level requirement of the motor, and the maximum temperature of the permanent magnet was 65.6 °C, which did not exceed the temperature safety value of the permanent magnet and ensured the accurate calculation of the component loss and motor

temperature. However, to reduce the energy losses, the wheeled motor was not properly cooled down. The literature [22] offers an analysis of the hazards of electrical heating for the transmission of radio energy with power up to the kW level. A simulation model was developed under current conditions of several hundred amperes at kilohertz, and the loss of energy was analyzed mainly in the form of heat, and it was found that the heat loss was mainly related to the environment and the heating of the cable conductors. Due to the presence of more energy in the air, the temperature raised very quickly, and the collector cable caused significant changes in temperature in a short time, causing aging problems in the cable line. The calculation method of core loss and winding loss in the design of forced air convection high-frequency transformers has been presented in [23]. Analytical models of winding losses for multilayer copper foil and Leeds wire windings in the presence of magnetic forces were developed. To verify the accuracy of both models, they were selected for comparison with two-dimensional finite element simulations, with errors within 1% and between 5–9% for the copper foil and Leeds wire models, respectively. The amount of data obtained was not large enough to make a comparison. In [24], radio energy transmission using power transfer technology was studied for a road inductive power transmission mat within a model pavement, which was analyzed for electromagnetic-thermal coupling, and the charging plate located on the ground showed power losses, caused by local temperature rise problems. Numerical thermal analysis of the double-D inductive power transfer (IPT) mat was proposed, and the mat was placed flat on the road surface and buried in the model road surface. The temperature can reach 100 °C when placed flat on the pavement, and the temperature value was 87 °C when the transmission pad was buried. When the duty cycle was 0.67, the temperature of the pad was 76 °C. The 5-min cooling time reduced the operating temperature of the IPT system. However, the temperature variation corresponding to multiple sets of different duty cycles was not taken into account. The object of study in [25] was a pickup for wireless energy transmission, which was first modeled and then analyzed the effect of the variation of resistance and capacitance on the efficiency of the system at different temperatures, and finally, the temperature of the device was reduced by using appropriate wires and capacitors to improve the power transmission and by using a heat sink. However, only one type of heat sink specification was used in the model. The literature [26] was a thermal analysis of an electric vehicle wireless energy transmission system under different misalignment conditions. During the wireless charging process, the receiver and transmitter are difficult to align perfectly, and in this case, the wireless energy transmission was carried out for a long period and the high temperature of the system raised safety issues. Thus, it was proposed to use a genetic algorithm. to predict the safety risk, that simulated the transient thermal distribution with JMAG software and established the temperature rating framework as the evaluation criteria. When the misalignment distance was greater than 81.12 mm, or the tilt angle was more than 6.83 °C, the highest temperature of 104.97 °C was reached. However, no cooling measures were proposed in the corresponding temperature class. The literature [27] was for situations when the static wireless charging system is operated for a relatively long period; the magnetic coupler has been heated up, which develops adverse effects on the stability and safety aspects of the system. Therefore, using simulation software to study the magnetic coupler heating, the temperature of the coil can reach 106 °C. Different methods, such as using cooling materials for the core layer and slight oxidation on the surface of the core layer, were used to reduce the temperature, and the experiment verifies that the cooling materials can effectively reduce the temperature to 91 °C. The disadvantage was that the influence of the core layer shape design on the heat dissipation capability was not considered. In [28] was a study of assemblies of interconnected, stacked flexural substrates to investigate the effects of the substrate thermal conductivity, potted adhesive thermal conductivity, and air convection coefficient on the thermal characteristics of interconnect assemblies, and the optimal chip layout scheme of interconnect assemblies was obtained after comparison to make its overall temperature more appropriate. However, there is a lack of practical models for verification.

Most of the above studies were aimed at optimizing the thermal layout of stacked 3D chip components. In the studies of foreign object detection, electromagnetic field distribution characteristics, and the effect of current magnitude on the temperature of the battery in magnetic resonant wireless charging, the heat generated by the research object can be effectively reduced by changing the structural design and physical heat dissipation, so that the system performance can reach the optimum state. However, the above studies consider a single way of cooling and did not consider the influence of material factors on the overall temperature of the system. Therefore, this paper proposed to study the influence of material factors on the temperature level of the wireless charging system and its distribution.

In the wireless charging system, most of the electrical components produce different degrees of electrical energy loss, so the temperature analysis and calculation of the coils in it were needed in the wireless charging system of electric vehicles. In this paper, we study the magnetically coupled coil of an electric vehicle wireless energy transmission system. Based on the study of the basic laws of heat transfer, we simulated a realistic scenario of an electric vehicle, built a coil model using ANSYS software, set boundary conditions and relevant parameters, and analyzed the temperature distribution on the coil surface. It is hoped that this paper will provide a reference for the design of magnetically coupled resonant coils.

## 2. Basic Theory of Heat Transfer

The electric vehicle wireless charging mechanism mainly consists of transmitting and receiving coils, etc. Its temperature field distribution is a relatively complex heat conduction problem. Based on the basic theory of heat transfer and fluid mechanics, we analyze and study the heat transfer generated in its working condition and combine the law of conservation of energy to calculate the temperature rise around the coil. There are three ways to perform heat transfer, namely, heat conduction, convection heat transfer, and radiation heat transfer.

The phenomenon of heat conduction caused by temperature differences in the same medium or between non-identical media becomes heat conduction. The basic law of heat conduction was derived from Fourier's law, which means the amount of heat flowing past a given area in a unit of time. The heat flow rate of conduction is proportional to the temperature gradient and the cross-sectional area perpendicular to the direction of heat conduction. The expression is:

$$Q_1 = -\lambda A_1 \frac{\mathrm{d}t}{\mathrm{d}x} \tag{1}$$

where $Q_1$ is the heat transfer heat flow (W); $\lambda$ is the thermal conductivity of the material (W/(m. °C)); $A_1$ is the cross-sectional area in the vertical thermal conductivity direction ($m^2$); $\frac{\mathrm{d}t}{\mathrm{d}x}$ is the temperature gradient along the isothermal surface normal to the direction (°C/m). The negative sign in the formula indicates that the direction of heat transfer is opposite to the temperature gradient. It can be seen that if you want to increase the amount of heat dissipation by heat transfer, you can increase the thermal conductivity and choose a material with high thermal conductivity. In general, the thermal conductivity of metals is relatively high, followed by non-metals; liquids are relatively low, and gases are the smallest.

Convective heat exchange is the process of heat exchange that occurs due to the difference in temperature, a process that takes place between a flowing fluid and the solid surface it contacts. According to the main factors that affect its change, such as density difference and external pressure, convection heat transfer can be divided into two types. The first is natural convection, which is caused by the difference in the density of fluids with different temperatures, while the second is caused by the external pressure forcing the

fluid to undulate, which becomes forced convection, in other words, because of the flow caused by the pressure difference. The formula of convective heat transfer is:

$$Q_2 = h_c A_2 (t_w - t_f). \tag{2}$$

In Equation (2), $Q_2$ is the amount of convective heat exchange (W); $h_c$ is the convective heat exchange coefficient (W/(m. °C)); $A_2$ is the effective convective heat exchange area of the contact surface, and its unit is ($m^2$); $t_f$ is the temperature of the cooling fluid, and its unit is (°C); and $t_w$ is the temperature of the solid surface, and its unit is (°C). From Equation (2), it is clear that to enhance convective heat transfer, what can be considered is to increase the convective heat transfer coefficient as well as the convective heat transfer area. For natural and forced convection heat transfer, the equation presents the criterion equation for calculating the convection heat transfer coefficient and then calculates its associated coefficients according to different criterion equations, and this treatment can be applied to the Icepak module in ANSYS software for heat dissipation calculation.

When an object transmits energy to the outside world using electromagnetic waves, such a process is known as radiative heat transfer. In any object above absolute zero, it radiates energy to the outside world at a certain wavelength, and at the same time receives the energy radiated to it from the outside world. The thermal radiation between objects is mutual, and if there is a temperature difference between objects, the process of thermal radiation between them will take place. The formula for calculating the radiation heat exchange between the surfaces of two objects is:

$$Q_3 = \delta_0 A_3 \varepsilon_{xt} F_{12} \left( T_1{}^4 - T_2{}^4 \right), \tag{3}$$

$$\varepsilon_{xt} = \frac{1}{\frac{1}{\varepsilon_1} + \frac{1}{\varepsilon_2} - 1}. \tag{4}$$

In Equation (3): $Q_3$ is the convective heat exchange rate (W); $\delta_0$ is the Stephan-Boltzmann constant, $5.67 \times 10^{-8}$ W/($m^2.K^4$); $A_3$ is the surface area of the object radiative heat exchange; $\varepsilon_{xt}$ is the system emissivity, where $\varepsilon_1$, $\varepsilon_2$ are the surface of the high-temperature object and the low-temperature object; $F_{12}$ is the angular coefficient from surface 1 to surface 2; and $T_1$ and $T_2$ are the absolute temperatures (K) of surface 1 and surface 2. It can be seen from the formula that to increase the radiation heat exchange between the object surfaces, the emissivity of the heat source surface, the angular coefficient from the hot surface to the cooling surface, and the surface area of the increased radiation heat exchange can be increased.

## 3. Thermal Simulation Analysis

With the rapid development of modern science and technology, computer-aided engineering analysis and design technology, mainly finite element technology, has been widely used. The platform for building the model in this study was ANSYS software, a wide commercial suite of engineering analysis software, which is the leading engineering simulation technology integration platform in the industry sector, with very powerful functions and the involvement of more application areas, such as structural, fluid, thermal, electromagnetic, and their mutual coupling analysis. The ANSYS software is highly open and flexible and provides solutions for every phase of product design, including general physical field numerical simulation, industry analysis, and model building. Design analysis, multi-objective optimization, and customized structural analysis solutions are also available.

### 3.1. Simplification and Import of the Model

Figure 1 is a schematic diagram of the wireless charging system for electric vehicles. As can be seen in Figure 1, the transmitter device is placed below the ground, which makes it difficult to dissipate heat due to poor air circulation, which can easily lead to

heat accumulation, temperature rise, damage to the device, and difficulty in maintenance. The receiving coil is installed in the lower part of the vehicle, which requires waterproof, dustproof, and other encapsulation protection treatments, leading to difficulties in its heat dissipation and, therefore, requiring thermal analysis and design to ensure its operational reliability. The transmitting coil and receiving coil are the main components of the electric vehicle radio energy transmission system. The transmitter coil generates an alternating magnetic field under the action of an alternating current, and the receiver coil generates an induced electric potential under the action of the magnetic field for energy transmission. To simulate the electric vehicle radio energy transmission system, it is necessary to first simplify the radio energy transmission system, combine the simulation features of the ANSYS software platform, and transform it into a thermal model that can be solved computationally while keeping the basic structure and state unchanged. There are two methods of building the thermal model: one is to use the component module that comes with the ANSYS software for modeling, and the second is to use an external CAD model import. In this study, we used the software's component module for modeling. When using Icepak for thermal simulation, geometric features that do not affect heat dissipation need to be deleted during the model import process, while geometries that are not suitable for the software for thermal simulation must be modified and simplified. The model was imported into ANSYS through the Geometry module and modeled and analyzed in the ANSYS software platform. The flowchart of the thermal analysis performed in this study is shown in Figure 2.

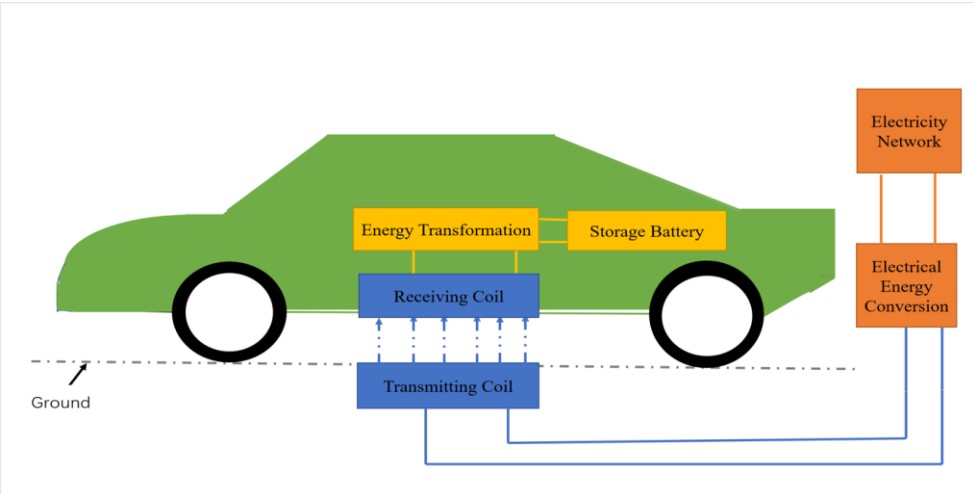

**Figure 1.** Schematic diagram of the wireless charging system for electric vehicles.

*3.2. Boundary Conditions and Parameter Settings for the Solution*

The simulation analysis was carried out in the Icepak module of ANSYS analysis software, which was divided into four steps: parameter setting, dividing the network, calling the solver to solve, and post-processing display. The parameter set included the boundary conditions and the basic settings of the solution.

The simulation process also required parameter setting for the thermally simulated, wireless energy transmission system, where the most important parameters were the electrical conductivity, thermal conductivity, heat, and specific heat capacity of copper, ferrite, and bakelite in the coil structure, which directly affect the accuracy and precision of the simulation. Theoretically, it is necessary to model the coil according to its actual physical structure, and then assign actual materials to the coil, housing, and core, respectively.

Where the solution domain range was set, Icepak requires that the solution domain is large enough to ensure that the boundaries do not affect the calculation results. The maximum dimension of the model in three directions was 100 mm, which is denoted by L. The values of the boundary condition parameters are shown in Table 1, and the simulation diagram of the set boundary conditions is shown in Figure 3.

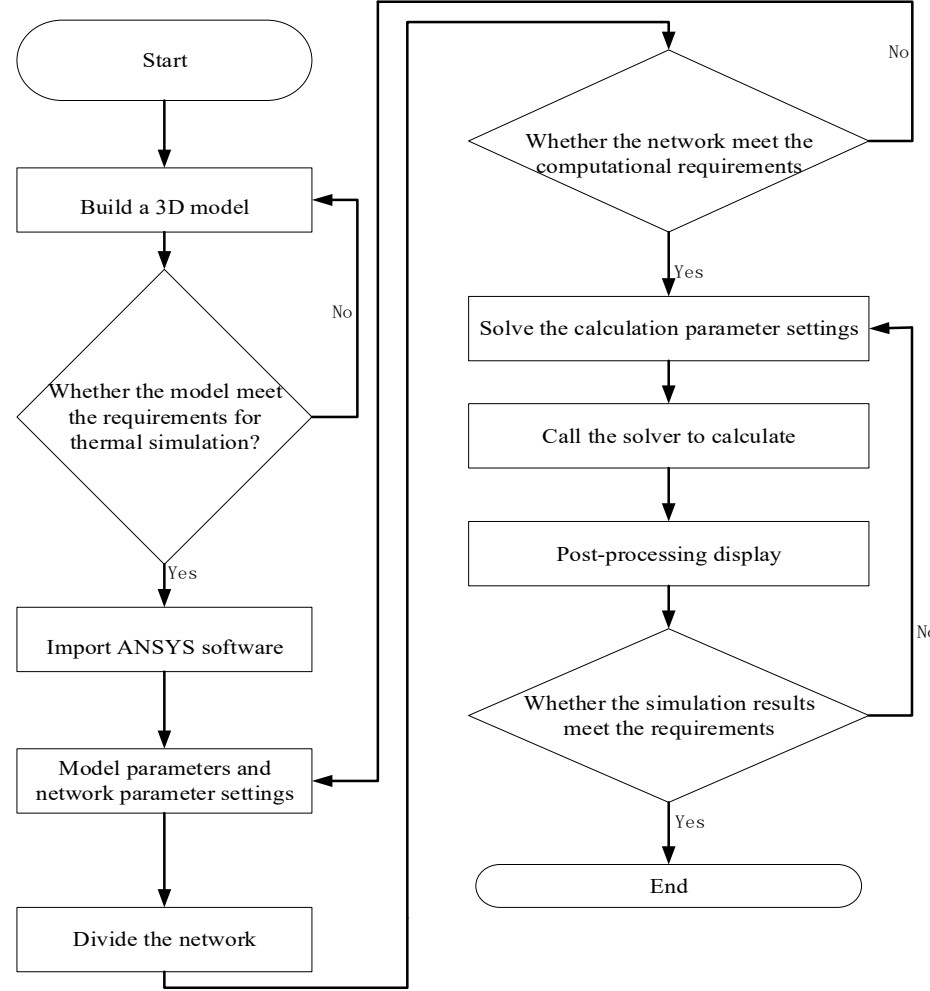

**Figure 2.** Thermal analysis flow chart.

**Table 1.** Boundary condition parameter setting diagram.

| Name | Value | Evaluated Value |
|---|---|---|
| Command | CreateRegion | / |
| Coordinate System | Global | / |
| +X Padding Type | Percentage Offset | / |
| +X Padding Data | 300 | 300 |
| -X Padding Type | Percentage Offset | / |
| -X Padding Data | 300 | 300 |
| +Y Padding Type | Percentage Offset | / |
| +Y Padding Data | 300 | 300 |
| -Y Padding Type | Percentage Offset | / |
| -Y Padding Data | 300 | 300 |
| +Z Padding Type | Percentage Offset | / |
| +Z Padding Data | 600 | 600 |
| -Z Padding Type | Percentage Offset | / |
| -Z Padding Data | 300 | 300 |

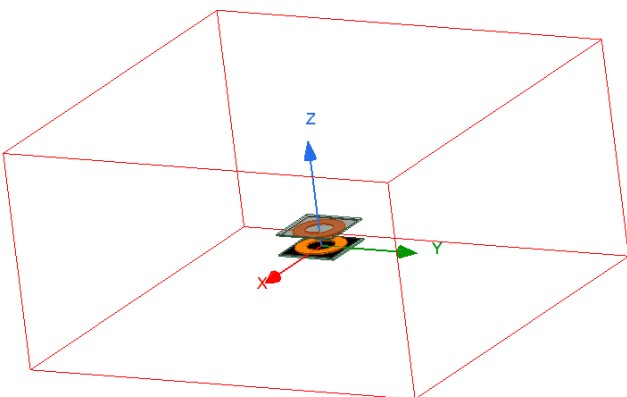

**Figure 3.** Simulation diagram of setting boundary conditions.

Table 1 mainly sets up the calculation area of the coil model and sets up the calculation side lengths for the six directions of the X-, Y-, and Z-axes. According to the requirements of the software for setting the thermal stability model, the upper distance of the model (the opposite direction of gravity) should be at least 2 L, the minimum value on the left and right sides should be 0.5 L, and the distance in the direction of gravity should be at least 1 L. In this example, the receiving coil was installed on the car, and the space was limited, so the X-axis, Y-axis positive and negative directions, Z-axis negative direction 3 L, and +Z direction 6 L were set. Figure 3 shows a diagram of the model after setting up the calculation area. The red bounding box range is the calculation area for the visualization effect. The orange part of the figure is the coil, and the black part of the figure is the core layer, which is divided into two parts, namely the transmitting coil and the receiving coil. In the receiving part, the receiving coil was very clear due to the transparency parameter of 0.5.

### 3.3. Dividing the Grid

After the basic parameters and boundary conditions were set, the thermal model needs to be meshed. The mesh division was an important part of the numerical calculation and simulation, and the quality of the mesh division can directly determine the accuracy and speed of the solution calculation. On the one hand, a too-dense partition would lead to a multiplication of the number of meshes, resulting in low computational efficiency and difficulties with convergence. On the other hand, a sparse network degrades the overall network quality and affects computational accuracy. Therefore, using discontinuous networks to locally encrypt regions with poor grid quality was the most common approach to using Icepak.

The grid division is divided into three steps: (1) division of the system grid; (2) grid encryption; and (3) grid quality adjustment. The overall discontinuous grid was used in Figure 4, the multi-level network was used internally, and the structured grid was used externally, which can reduce the number of grids and make the calculation easy to converge. After the delineation was completed, the grid was checked to ensure there was no distortion and the calculation could be solved.

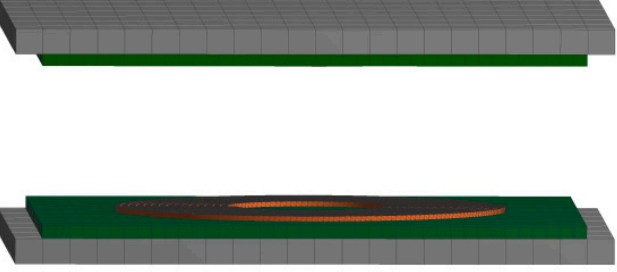

**Figure 4.** Model after meshing.

### 3.4. Modeling and Parameter Setting

The model built in this study was for the simulation and analysis of the power transmission system of a coupled resonant wireless charging car. The appearance of the model is shown in Figure 5. The coil material in the model is copper wire, and the material of the external protection shield is made of aluminum metal. In Figure 5a, the inner radius of the coil is 50 mm, the growth radius is 4.05 mm, the number of turns of the coil is 10, the transmitting and receiving coils are equal, the offset distance is 0 mm, and the relative distance between the two coils is 60 mm. In Figure 5c, the cylindrical spiral coil has 10 turns with a wire cross-section of 2.05 mm × 2.05 mm and an initial radius of 50 mm.

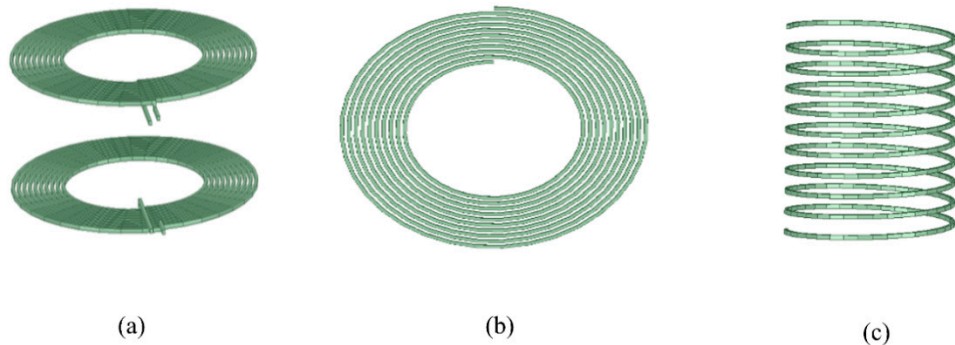

(a)         (b)         (c)

**Figure 5.** Coil model diagram. (**a**) Two coil model, (**b**) Single coil model, (**c**) Cylindrical spiral coil model.

Considering the proximity effect between the multi-turn coils, it is necessary to set the distance of mutual proximity in the multi-turn coils to 0 mm, replace different materials, and adjust the heat transfer parameters, while the other parameters remain unchanged, as shown in Table 2.

**Table 2.** Material parameters.

| Materials | Relative Magnetic Permeability | Electrical Conductivity (S/m) | Thermal Conductivity (W/(m·K)) | Specific Heat Capacity (kJ.kg$^{-1}$.$K^{-1}$) |
|---|---|---|---|---|
| Copper | 1 | $5.8 \times 10^7$ | 401 | 0.39 |
| Bakelite | 1 | $1 \times 10^{-10}$ | 0.8 | 0.82 |
| Aluminum | 1 | $3.56 \times 10^7$ | 237 | 0.90 |
| Gold | 1 | $4.56 \times 10^7$ | 314 | 0.13 |
| Graphite | 1 | $1.00 \times 10^5$ | 24 | 0.71 |
| Iron | 1 | $1.04 \times 10^7$ | 79.5 | 0.45 |
| Silver | 1 | $6.30 \times 10^7$ | 429 | 0.23 |

Figure 6a shows the top view of the coil model, in which the maroon circle is the coil, which is made of copper. The green square in the figure is the core layer and is partially covered by the circle, where the outermost gray area is the protective layer, made of Bakelite. The parameters of the coil model were set accordingly by reviewing the literature and considering the very limited installation space in the car. The coil had a length, $L_1$, of

220 mm, a width, $W_1$, of 210 mm, an inner diameter, $\Phi_2$, of 100 mm, and an outer diameter, $\Phi_1$, of 181 mm. Figure 6b shows the side view, divided into two parts—the upper and lower parts—with a distance $h$ of 60 mm. The upper part was placed on top of the electric car body and the lower part was mounted on the ground for energy supply. The yellow color represents the coil, the green color represents the core layer, and the remaining gray part was used to protect the wireless charging system from dust and water.

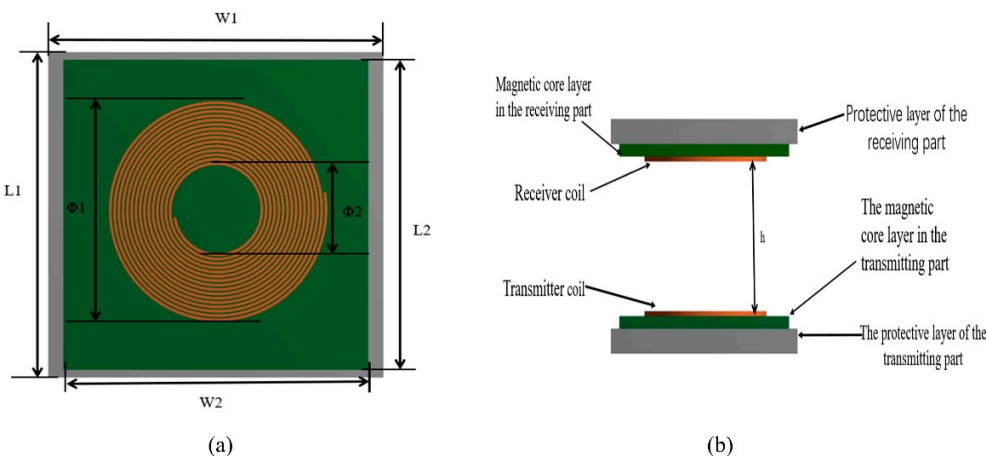

(a)                                           (b)

**Figure 6.** Coil model view. (**a**) Top view of the coil model Figure, (**b**) Side view of the coil model.

## 4. Experiment and Analysis

In the simulation process, the gap distance length of the model was 60 mm, and the electromagnetic field and temperature field simulations were performed. The electrical parameters of the coil can be obtained using the electromagnetic field, where the distribution of the magnetic induction intensity B is shown in Figure 7. Due to the presence of the iron core in the coil, the magnetic field was concentrated on the coil, and the loss values of each part of the model can be obtained, where the loss value of the transmitting coil was 17.05 W and the loss of the receiving coil was 16.06 W. The transmission efficiency of the coil was 90%.

Figure 7a,b show the magnetic field distribution of the two sub-coils. The upper part of the two figures shows the magnetic induction cloud of the receiving coil, and the lower part shows the magnetic induction cloud of the transmitting coil. The red color indicates a strong magnetic induction, the green color a moderate magnetic induction, and the blue color a small magnetic induction. The phenomenon in the figure indicated that the closer the part of the coil transmitting, the greater the magnetic induction, and when deviating from the part of the coil, the magnetic induction was smaller.

When the coil model was placed in the bakelite shell, its main heat transfer methods are convective heat exchange of air located in the shell, and convective heat exchange between the outer shell surface and the nearby air. The initial temperature of the environment was set to 22 °C, the materials of the core layer and coil in the coil model were changed, and the bakelite material was used for the protective layer. After the magnetic thermal coupling temperature was stabilized, the magnetic coupling temperature of the coil is shown in Figure 8.

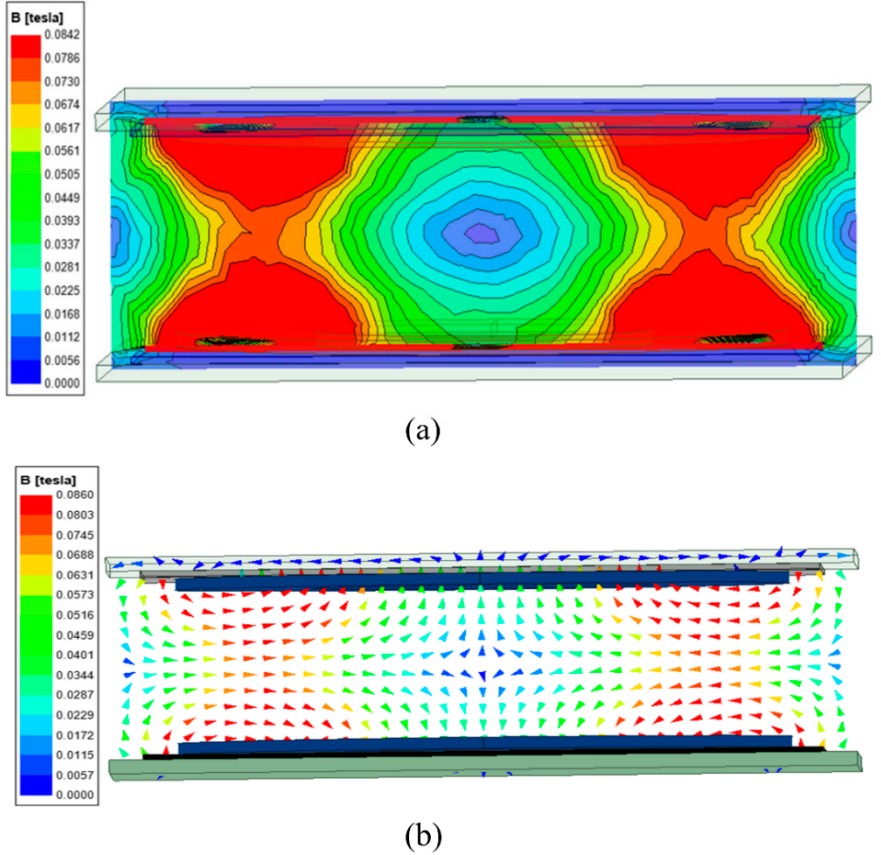

**Figure 7.** (**a**) Magnetic field distribution diagram, (**b**) magnetic field distribution (vectorial).

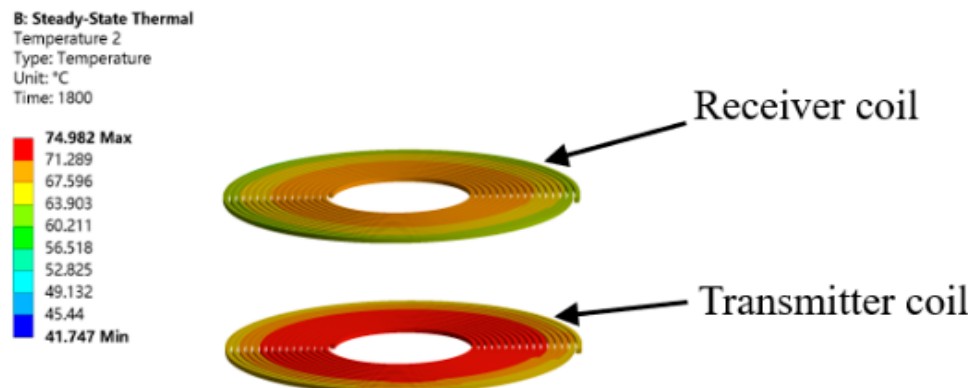

**Figure 8.** Temperature cloud of the coil section.

Figure 8 shows the temperature cloud of the coil part, and the material used for the coil is copper. As shown in the legend on the left side of the figure, the maximum value of the temperature is 74.982 °C and the minimum value of the temperature is 41.747 °C. The temperature of the transmitting coil is higher than that of the receiving coil, mainly because the receiving coil is under the action of the magnetic field, and the receiving coil does not fully receive the magnetic flux of the transmitting coil, making the temperature of the receiving coil lower than that of the transmitting coil. The temperature of the coil placed on the ground decreases from inside to outside, mainly because the middle of the coil is less in contact with cold air.

Figure 9 shows the temperature cloud diagram, divided into two parts, upper and lower, with the red dashed line in the diagram as the dividing line. Figure 9a is composed

of the coil and the core layer. Figure 9b is made by adding a protective layer on top of the upper part.

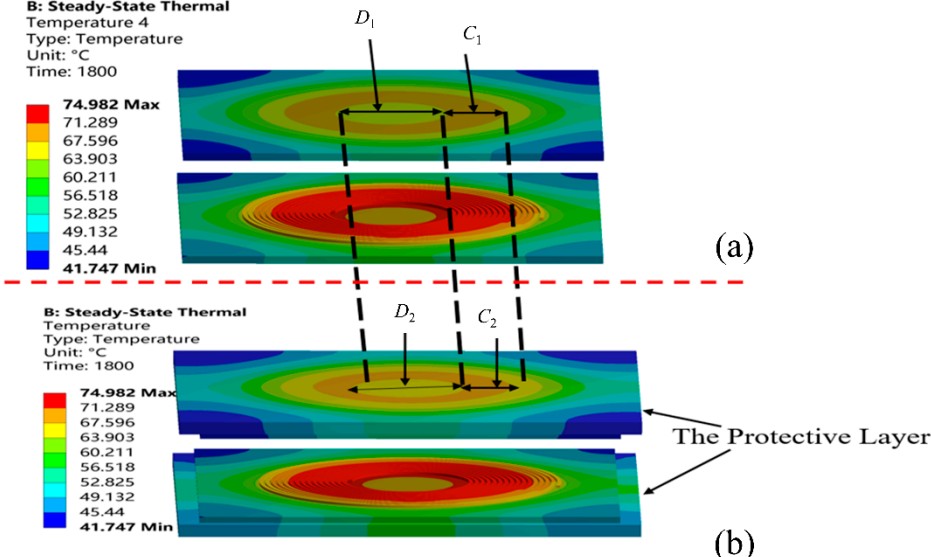

**Figure 9.** Temperature clouds (**a**) temperature clouds of the core layer and coil layer, (**b**) overall temperature clouds of the coil model.

Figure 9a shows the overall temperature distribution of the coil model without the protective layer. The material used in the model coil is copper and the core layer is made of aluminum. The maximum and minimum values of temperature are the same as the range of temperature values for the coil without the core layer added, but the addition of the core layer causes the temperature distribution to change, and the corresponding temperature range becomes larger in the area of the graph.

Figure 9b shows the temperature cloud of the coil model after adding the protective layer. Compared with Figure 9a, the temperature maximum value of 74.982 °C and the minimum value of 41.747 °C do not change. The area of the area referred to by $D_2$ in Figure 10 is larger than the area of the area referred to by $D_1$, and the area of that referred to by $C_2$ in Figure 9 is smaller than the area range referred to by $C_1$.

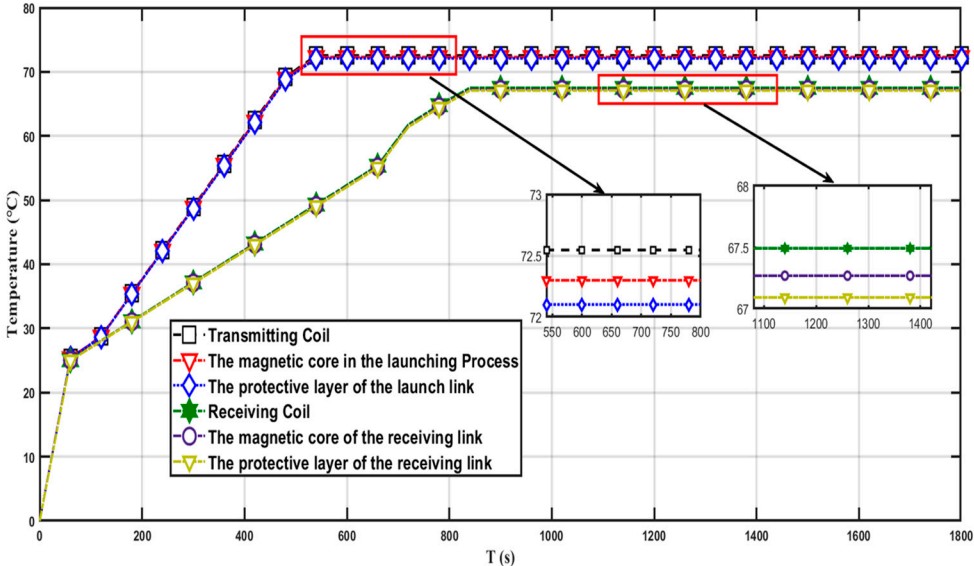

**Figure 10.** The temperature profile of the coil model with different materials.

The temperature range of the area specified by $D_1$ and $D_2$ is from 63.903 °C to 67.596 °C. The temperature of the area specified by $C_1$ and $C_2$ fluctuates from 67.596 °C to 71.289 °C. After adding the protective layer, the area where $D_1$ is located increases to $D_2$, and the area where $C_1$ is located decreases to $C_2$. The overall temperature of the receiving section decreases.

Figure 10 shows the temperature profile of the coil model with bakelite as the protective layer, aluminum as the coil, and copper as the core. As a whole, the temperature of the transmitting part of the model is higher than that of the receiving part. Between 0 and 500 s, the temperature of the transmitting part appears to increase sharply and then smoothly as time increases. In the time from 500 to 1800 s, the temperature of the transmitting part tends to a value that is stable and constant around 72.5 °C. The temperature difference of the transmitting part, which consists of the protective layer, the core layer, and the transmitting coil layer, is relatively small from the local magnification on the figure, and the difference is not more than 0.5 °C for these three layers. The temperature of the receiving part increases between 0 and 900 s with increasing time, and the temperature rises quickly and then slowly. The temperature of the receiving part no longer changes significantly from 900 to 1800 s but is stable at around 68 °C. The local plot in the figure shows that the maximum temperature of the transmitting coil is around 72.5 °C and the maximum temperature of the receiving coil is 67.5 °C.

Figure 11 shows the temperature profile for fixing the material of the coil and changing the material of the core layer. When the charging system model is operated for 30 min, it is clear from the graph that when the core layer material is aluminum and six different materials are used for the coils, the highest temperature is using graphite, followed by iron. The lowest temperatures are silver and copper. Considering the production cost and demand, copper material is the most suitable choice. When the core layer is graphite, for the coil using six materials, the temperature of the coil is over 74 °C, the highest temperature reaches 87 °C, which did not meet the temperature safety standards. Since graphite is brittle and difficult to bend, it was not conducive to making a magnetic core layer.

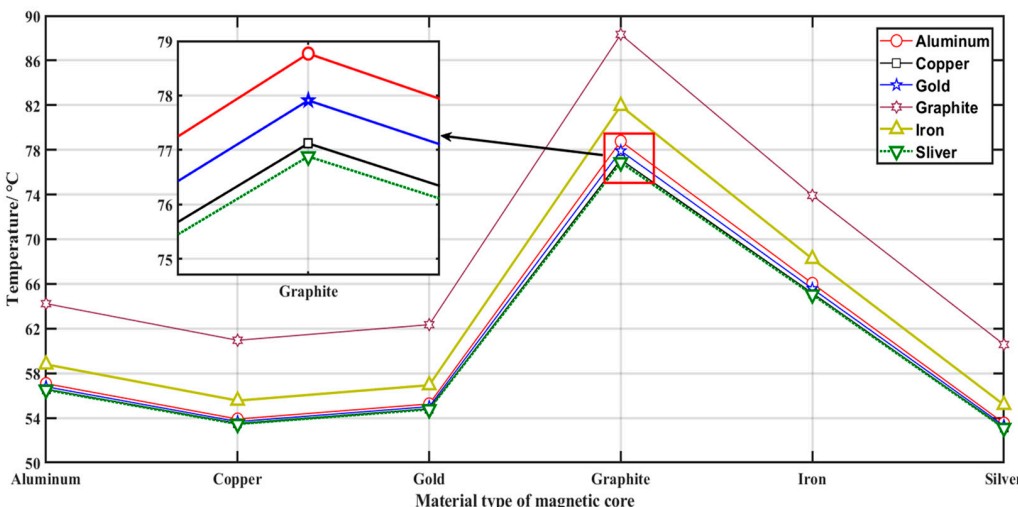

**Figure 11.** Temperature profiles for different core materials.

The temperature profile of the plate with bakelite as the protective layer of the system, copper as the coil, and aluminum as the core is shown in Figure 12. The graph shows that the temperature of the transmitting coil increases with increasing time value in the range of 0 s to 480 s. Between 480 s and 1800 s, the temperature of the firing coil tends to a constant value of 71.5 °C.

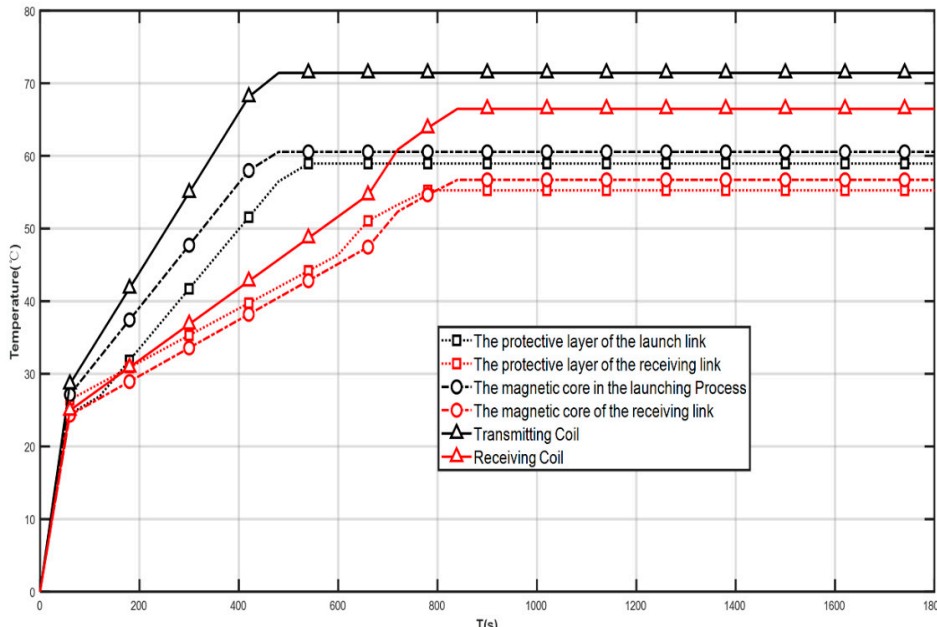

**Figure 12.** Plate layer temperature profile.

The temperature of the protective layer and the core layer in the firing section shows an increasing trend from 0 s to 500 s. In the time interval from 500 s to 1800 s, the maximum temperature of each layer in the firing section, except the firing coil, does not exceed 60 °C.

The temperature of the protective layer, core layer, and receiving coil layer in the receiving section increased with time from 0 to 840 s. In the range of time values of more than 840 s and less than 1800 s, the temperature of each layer in the receiving section no longer changed significantly, and the maximum temperature of the receiving coil was 66.5 °C. The maximum temperature of the core layer in the receiving section is 56.7 °C, and the temperature of the protective layer is 55.25 °C. The maximum value is 55.25 °C.

When the coil is made of copper and works for 30 min, the highest temperature in the system is the transmitting coil, and its maximum temperature is 74.982 °C, which is in line with the normal charging temperature of wireless electric vehicles. According to the general design requirements, the civil grade working temperature range is 0–90 °C, the material is selected from domestic grade original parts, the process treatment is waterproof, and the circuit board is integrated. The military-grade working temperature range is −55–125 °C, possesses auxiliary circuit and backup circuit design, shock resistance, high-temperature resistance, and multiple protections. The coil temperature studied in this research was within its normal range, and the transmission efficiency was high from the perspective of thermal simulation.

In the simulation calculation, natural convection heat dissipation at an ambient temperature of 22 °C is simulated. Figure 10a shows the temperature distribution of the coil cross-section, from which it can be seen that the energy is mainly concentrated on the coil surface in the coil structure. The coil temperature variation is more significant, and the core temperature is higher than the surface temperature. Another reason was the magnetic coupling loss, as the coil heat dissipation using natural convection, resulting in heat, is not easily dissipated, the system level (Bakelite layer) on the overall temperature impact is small to negligible. As can be seen in Figure 8, the overall transmit and receive coils have a uniform temperature distribution in the cavity.

The airflow velocity parameters between the transmitting and receiving sections are set, and the state and velocity distribution of the external fluid flow are shown in Figure 13. Figure 13a shows the setting of the wind speed. Figure 13b shows the effect on the velocity coil model, in which the red color indicates the maximum value and the blue color indicates the minimum value. The air inside the coil flows upward as a whole, and the flow velocity

is higher in the lower part, and the maximum velocity occurs between the transmitting and receiving coils, with a maximum velocity of 0.52 m/s. Significantly, the highest rate is consistent with the highest temperature, and the rate is also smaller in the area below where the heat flow density is smaller; thus, the reliability of the experiment and the correctness of the theory is verified.

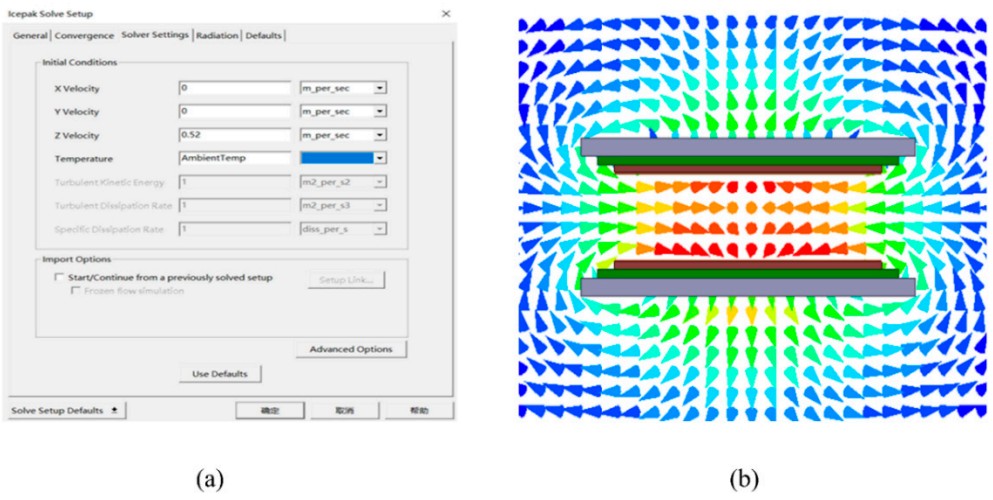

(a)    (b)

**Figure 13.** Speed setting diagram (**a**) Speed setting chart (**b**) Speed effect chart.

In this study, the calculation model was chosen as an eddy current model, and the model solution was undertaken in the ANSYS software. Figure 9a shows the temperature distribution of the transmitting and receiving coils at the maximum temperature of 74.982 °C and 71.289 °C for the magnetically coupled resonant transmitting and receiving coils. As seen in Figures 8 and 9a, the temperature on the transmitting and receiving coils is higher outside the coil heat transfer efficiency, and it is more uniformly distributed on the coil. From Figures 10 and 12, it can be seen that the coil temperature grows with time, showing a fast and then a slow rate, and finally, the coil temperature tends to be stable, which is in line with the actual situation. The coil temperature rises faster at the beginning of a period, and after 13 min, the coil temperature rise is basically unchanged, which is generally consistent with general common sense and in line with the temperature range of normal operation of electric vehicles, and hopefully provides a valuable reference for studying electric vehicle coils and improving thermal simulation work.

## 5. Conclusions

Coil thermal analysis is an important element in the design of the wireless charging system for electric vehicles. In this paper, ANSYS software was used to build a coil model to simulate the temperature distribution on the coil surface when the electric vehicle wireless charging system was in normal use. After simulating and comparing the results, the transmitting coil and receiving coil were finally determined to be equal in size. The inner diameter of the coil was 100 mm, the outer diameter of the coil was 181 mm, and the coupling distance between the transmitting and receiving coils was set to 60 mm. Six materials were used for the coil part of the electric vehicle wireless charging system, and the established model was allowed to run for 30 min. From the point of view of temperature parameter range, gold, silver, copper, and aluminum materials were used to make the coils. The coil material was finally selected to be copper, based on the combination of the cost of obtaining the material, the ease of production, and the performance. When copper was used to make the coil, the maximum temperature of the coil was 74.982 °C, and the temperature distribution on the coil gradually decreased from inside to outside, which is in line with the normal heat dissipation situation. It is hoped that this study will provide a reference for the design of wireless charging coils for electric vehicles and can help promote and develop the application of wireless charging systems for electric vehicles.

**Author Contributions:** Conceptualization, Q.X. and C.W.; funding acquisition, Q.X., C.W., X.Z., Y.L. (Yuanxiong Liang) and K.L.; methodology, M.C., X.Z. and J.W.; resources, M.C.; software, C.W. and Q.X.; supervision, Y.L. (Yunyun Luo); validation, Q.X. and C.W.; visualization, Z.X.; writing—original draft preparation, C.W. and Q.X.; writing—review and editing, C.W., M.C. and Q.X. All authors have read and agreed to the published version of the manuscript.

**Funding:** This research was funded by the Guangxi Key Laboratory of Power System Optimization and Energy Technology.

**Acknowledgments:** The authors are very grateful to the reviewers for their care in their unbiased and constructive reviews, which have greatly assisted the revision of this manuscript.

**Conflicts of Interest:** The authors declare no conflict of interest.

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
