# Peer review of "Thermal Analysis of Coupled Resonant Coils for an Electric Vehicle Wireless Charging System"

_wevj, doi:10.3390/wevj13080133_

Round 1

Reviewer 1 Report

This paper focuses on the thermal analysis of a wireless energy transfer system to charge an electric vehicle. The aim is mainly concerned on the causes of degradation, that excessive temperature may cause on the operating hardware. An extensive literature review was undertaken, to show which are the present state-of-the-art systems and to place the present work among them. A simulation model was devised, based on the heat transfer theory, such that the main components of the wireless energy transfer system could be included. Both the transmitting and the receiving ends of the wireless system are considered. The set of accounted components encompasses the coils, the magnetic core layers, and the protective layers. According to the simulations carried out, the temperature profile was studied, namely through the variation of the material used to build the coil. A valuable and interesting work has been carried out. The theme is interesting, the paper is well organized, and the idea is easy to follow, although some details should be made clearer. The quality of the written English is good.

However, there are some remarks regarding the content of the paper, to strengthen its content. The main issues are concerned about clarifying some points, and correcting some few typo mistakes, as follows:

Mark 51 – “perform” should be inserted between “efficiently” and “wireless”.

Mark 101 – The symbol “HMCPL” should be defined, prior to its use in the text.

Mark 133 – The symbol “3D-MCM” should be defined, prior to its use in the text.

Marks 152 to 158 – The paragraph should be reformulated, as the key idea isn’t made very clear. Besides, the whole text is in one sentence alone, which is not a good practice.

Mark 160 – The second occurrence of “of the system” should be removed.

Marks 162 to 170 – The paragraph should be reformulated, as the key idea isn’t made very clear. Besides, the whole text is in one sentence alone, which is not a good practice.

Mark 290 – The symbol of “Watts” should be a capital W, as well as in every other occurrences in the text.

Mark 292 – The exponent “2” should be in superscript.

Marks 311 and 329 – The character “^” should be eliminated and exponents should be in superscript.

Mark 411 – The “a”, following “too dense”, should be before it.

Mark 421 – “Multi-Level” should be removed.

The subfigures of Fig. 6 should be properly identified with a), b) and c).

Mark 439 – The sentence shouldn’t end, but instead continue with “…, as shown in Table 1.”.

In Table 1, the column regarding the electrical conductivity, should present values where the character “^” should be eliminated and exponents should be in superscript.

The subfigures of Fig. 7 should be properly identified with a), and b). Regarding this figure, what were the main reasons for having selected the dimensions that were indicated? Please clarify.

Mark 460 – The space character following “mm” should be removed.

The subfigures of Fig. 8 should be properly identified with a), and b).

Mark 503 – “The upper part is Fig. a,” should be removed.

Mark 504 – “The lower part is” and “which” should be removed.

Mark 571 – The “And” before “military grade”, should be removed.

The subfigures of Fig. 14 should be properly identified with a), and b).

Mark 595 – Capitalization of the “And” before “reliability”, should be removed.

Mark 611 – “the” should be inserted between “of ” and “wireless”.

Mark 612 – “ANSYSR” should be replaced with “ANSYS”.

Mark 617 – “six” should be capitalized.

Reviewer 2 Report

There are numerous changes to be implemented for the acceptance of the manuscript. Kindly note the following

1-In the introduction the authors can provide the need for EVs for a sustainable future. Kindly refer to the following manuscript

Chakraborty, S., Kumar, N.M., Jayakumar, A., Dash, S.K. and Elangovan, D., 2021. Selected Aspects of Sustainable Mobility Reveals Implementable Approaches and Conceivable Actions. Sustainability, 13(22), p.12918.

2-Express kiloWatt as kW, also check all the abbreviations, SoC not SOC, follow the standard universal template

3-No need to provide the fundamentals of heat transfer, its available in high school books

4- Table 1 check the unit of sp. heat capacity its kJkg-1K-1

5-The level of the English language must be checked throughout the manuscript to make sure that the article is free from grammatical mistakes. Also, enhance the clarity of all the figures and adhere to the journal format

Reviewer 3 Report

The following specific remarks and concerns can be highlighted.

 1. The phrase “Therefore, the thermal analysis simulation of the electric vehicle wireless charging system to understand the heat distribution is beneficial to the effective cooling of the system, so that the system can efficiently wireless energy transmission.” is difficult to be understood and needs to be rephrased.

2. In the phrase “For the heat generated by electronic products, in this technology of controlling heat, foreign countries are more mature” the notion of “foreign” doesn’t make sense since the journal is an international one.

 3. At line 76, the notion of “hairiness” is inappropriate in the context of the phrase.

 4. A period (full stop) is missing at line 77, between the words “requirements High-density”.

 5.  At line 101 define the acronym “HMCPL”, i.e., High Magnetic Coupling Passive Loop.

 6.  The phrase “Literature [14] for lithium battery temperature estimation problem, considering the influence of battery aging on the battery temperature based on the battery structure and battery heat transfer law, the temperature of the battery surface is estimated using the finite element difference method, when the discharge rate reaches 0.5 C and 1 C's, the highest value of the internal temperature of the battery is 28.15 ℃ and 32.58 ℃, respectively is too long and contains six times the word battery.

7. The phrase “Literature [15] on the processing of long stainless steel tubes in the process of  magnetic field generator temperature rise is relatively large, more than the allowable temperature of the winding, inducing magnetic honing process is limited such a problem, in  order to ensure the sustainability of the process, so the cooling method of external circulation water cooling was proposed, simulation analysis and test results show that when 156 no water cooling circulation device is used, the temperature of the system maximum value 157 of 75 ℃. Is extremely long and confusing. There are several grammar mistakes and inconsistencies. Is it an automated translation?

8. At line 197 the symbol of “kilo” is “k”.

9. What do you mean by “… enters to raise safety issues.” at line 239?

10. The phrase “This paper focuses on the charging coils of the electric vehicle wireless charging electric energy transmission system, using thermal calculation and thermal simulation to perform thermal analysis and clarify their thermal distribution conditions for subsequent design.” Contains four times the word “thermal”.

11. In (1) the alpha symbol is in fact the differential symbol “d”. Quantity Q1 is in fact the heat transfer heat flow rate. The unit “watt” has the symbol “W” capital letter. Similarly, for Q2 in (2). It is a heat flow rate.

12. What do you mean by "software work" at lines 364 and 365?

13. The screen capture in Figure 3 is unnecessary, not providing any useful information.

 14. What do you mean by “What do you mean by the effect of the model?” at line 402?

15. Subfigures (a) and (b) are not indicated in Figures 8 and 14.

16. At line 491 correct “right” to “left”.

Round 2

Reviewer 2 Report

Make a comprehensive proof-reading 

Reviewer 3 Report

The authors have properly addressed my comments and made the necessary modifications or gave satisfactory explanations instead. There is a single correction to be made at line 306, namely to replace (??/??) with (dt/dt).
